# Syllogistic Reasoning for Legal Judgment Analysis

**Wentao Deng**[1], **Jiahuan Pei**[3], **Keyi Kong**[1], **Zhe Chen**[1], **Furu Wei**[2]
**Yujun Li**[1], **Zhaochun Ren**[4], **Zhumin Chen**[1], **Pengjie Ren**[1*]

[1]Shandong University, Qingdao, China. [2]Microsoft Research Asia, Beijing, China.
[3]Centrum Wiskunde & Informatica, Amsterdam, The Netherlands.
[4]Leiden University, Leiden, The Netherlands.
{wentao.deng, luxinyayaya, cz2021}@mail.sdu.edu.cn
{liyujun, chenzhumin, renpengjie}@sdu.edu.cn
Jiahuan.Pei@cwi.nl, fuwei@microsoft.com, z.ren@liacs.leidenuniv.nl

## Abstract

Legal judgment assistants are developing fast due to impressive progress of large language models (LLMs). However, people can hardly trust the results generated by a model without reliable analysis of legal judgement. For legal practitioners, it is common practice to utilize syllogistic reasoning to select and evaluate the arguments of the parties as part of the legal decision-making process. But the development of syllogistic reasoning for legal judgment analysis is hindered by the lack of resources: (1) there is no large-scale syllogistic reasoning dataset for legal judgment analysis, and (2) there is no set of established benchmarks for legal judgment analysis. In this paper, we construct and manually correct a syllogistic reasoning dataset for legal judgment analysis. The dataset contains 11,239 criminal cases which cover 4 criminal elements, 80 charges and 124 articles. We also select a set of large language models as benchmarks, and conduct a in-depth analysis of the capacity of their legal judgment analysis.

## 1 Introduction

Legal judgment assistants are growing rapidly due to large demands from legal practitioners and normal citizens (Cui et al., 2022) and the impressive progress of large language models (Liu et al., 2023; Li et al., 2021; Zhao et al., 2023). The core technique employed in those assistants is legal judgment analysis, which aims to analyze the description of a fact and produce reasonable results of a legal judgment.

There are two genres of approaches toward legal judgment analysis. One group of researchers is actively working on legal judgment prediction (LJP) (Feng et al., 2022). The main goal is to predict the results of a legal judgment (e.g., charges, legal articles, and terms of penalty) given the description of a fact (Xiao et al., 2018). This prediction

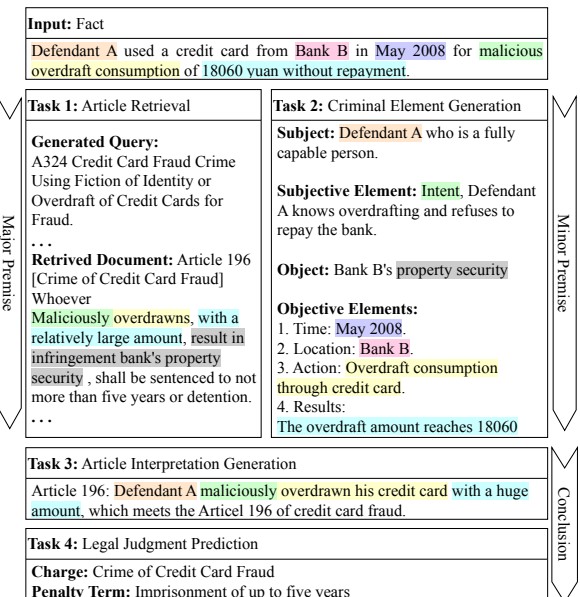

Figure 1: An example of criminal case of the credit card fraud. The output of task 1 is the main premise. The output of task 2 is the minor premise. The outputs of task 3 and 4 are the conclusion. The corresponding segments to different parts are marked in the same color. We mark the corresponding segments in different parts with the same color.

process is typically accomplished through multi-classification (Xu et al., 2020; Yue et al., 2021a; Feng et al., 2022) or generation (Deroy et al., 2023) based on LLMs. However, people cannot fully trust the results predicted by LJP without analytical details of legal judgments (Cui et al., 2022).

Another group of researchers explores court view generation (CVG) (Ye et al., 2018; Yue et al., 2021b; Wu et al., 2020) to enable the interpretability of the results of legal judgment. The core idea is to generate phrases or sentences in the description of a fact as rationales for supporting legal judgment results. However, the generated rationales might be specious (Chao et al., 2019) and can not represent the whole analysis process that leads to the final result.

---

*Corresponding author.

In practice, syllogistic reasoning is routinely employed by judges to assess the parties' arguments for legal judgments, thereby maintaining consistency and fairness in their decision-making process. (Hart et al., 2012; Constant, 2023). It allows judges to break down legal arguments into logical components, including major premise, minor premise and conclusion (Gardner, 2007; Constant, 2023).

In this work, we seek to answer the following research questions:

- *RQ1: How do LLMs perform on the syllogistic reasoning for legal judgment analysis?*
- *RQ2: How does syllogistic reasoning impact the performance of LLMs in legal judgment analysis?*

We make three main contributions to facilitate research into syllogistic legal judgment analysis (SLJA): a formalization, a dataset, and benchmarks. Let us expand on each of these.

We formalize SLJA as four interrelated tasks (See Figure 1): (1) article retrieval (AR) selects relevant articles from article database as a major premise; (2) criminal element generation (CEG) generates the criminal elements of a fact as a minor premise; (3) article interpretation generation (AIG) produces the applicable articles and corresponding rationale according to major and minor premises; (4) legal judgment prediction (LJP) generates the judgments results according to the applicable articles.

We construct a dataset for SLJA. First, we select criminal cases from the CAIL-Long dataset (Xiao et al., 2021) with the most common charges whose frequencies are larger than 40. Then, we invoke the ChatGPT API to generate the results of four tasks for the selected cases. Finally, we hire 22 workers with a legal background to revise the results generated by ChatGPT.

We select four state-of-the-art LLMs as benchmarks, including ChatGLM (Zeng et al., 2022), Davinci002, Davinci003, and ChatGPT [1]. We compare the performance of each model in each task with and without syllogistic templates (See Appendix A) as prompts. Through extensive experiments and analyses, we conclude that (1) SLJA is still challenging for state-of-the-art LLMs; (2) and syllogistic reasoning can provide effective information for LLMs to improve the performance of legal judgment analysis.

Our contributions can be summarized as follows:
- Four interrelated tasks proposed for syllogistic legal judgment analysis (SLJA). It is motivated by the practical application of syllogistic reasoning in legal judgments.
- The first Chinese SLJA dataset, which includes a detailed analysis process of syllogism. It facilitates the frontier researches on consistency and fairness in the legal decision-making process.
- Benchmarks based on the state-of-the-art LLMs, extensive experimental results, and in-depth analyses.

## 2 Related Work

### 2.1 Legal Judgment Prediction

Recent research focuses on exploring tailored tasks and datasets (See Table 1), and proposing effective models.

The task of LJP is originally defined to predict the results of a legal judgment given the descriptions of a fact (Kort, 1957). The early-stage focus was primarily on theoretical and quantitative analysis (Nagel, 1963; Keown, 1980), without large-scaled datasets. CAIL2018 (Xiao et al., 2018) is the first large-scale Chinese legal dataset for judgment prediction. Zhong et al. (2018) also includes annotations of the topological dependencies among articles, charges, and terms of penalty. These datasets have garnered significant attention due to their predictive results encompassing charges, penalty terms, and applicable law articles (Xiao et al., 2018; Chalkidis et al., 2019; Niklaus et al., 2021). Xiao et al. (2021) collect the cases with longer facts, which distribution is close to actual cases. To enable interpretability, Ge et al. (2021) provide annotations of article-fact correspondence for relevant law article recommendation. Wu et al. (2022) generate rationales for both charge and term of penalty, which are proved to be useful for LJP. An et al. (2022) annotate the criminal elements for each sentence in facts. However, none of these datasets can be used to comprehensively evaluate syllogistic legal reasoning when completing legal analysis tasks.

Traditional LJP models are focused on the statistical methods (Segal, 1984; Gardner, 1987). With the development of neural networks, RNN-based and transformer-based models have emerged as the prevailing models for LJP. Chalkidis et al. (2019) evaluate several GRU-based and BERT-based models and find that these models outperform the SVM

---

[1] https://platform.openai.com/docs/models/gpt-3-5

| Dataset | Task | | | | Statistic | | | | | |
|---|---|---|---|---|---|---|---|---|---|---|
| | AR | CEG | AIG | LJP | #CE Type | Avg. #Fact | #Charge | #Term | #Article | #Case |
| CAIL-2018 (Xiao et al., 2018) | ✓ | ✗ | ✗ | ✓ | - | 386 | **202** | 3 | 183 | **2,676,075** |
| CAIL-Long (Xiao et al., 2021) | ✓ | ✗ | ✗ | ✓ | - | 916 | 201 | **5** | **244** | 115,849 |
| DPAM (Wang et al., 2018) | ✓ | ✗ | ✗ | ✗ | - | **1,455** | 0 | 0 | 70 | 17,160 |
| TOPJUDGE-CJO[†] (Zhong et al., 2018) | ✓ | ✗ | ✗ | ✓ | - | - | 99 | 3 | 98 | 1,007,744 |
| TOPJUDGE-PKU[†] (Zhong et al., 2018) | ✓ | ✗ | ✗ | ✓ | - | - | 64 | 3 | 68 | 175,744 |
| TOPJUDGE-CAIL[†] (Zhong et al., 2018) | ✓ | ✗ | ✗ | ✓ | - | - | 122 | 3 | 105 | 113,536 |
| MLMN (Ge et al., 2021) | ✓ | ✗ | ✗ | ✓ | - | 146 | 0 | 0 | 86 | 1,189 |
| SCE (An et al., 2022) | ✗ | ✗ | ✗ | ✓ | 3 | 230 | 7 | 0 | 0 | 685 |
| RLJP (Wu et al., 2022) | ✓ | ✗ | ✗ | ✓ | - | 402 | 48 | 1 | 95 | 89,768 |
| Court-view-gen (Ye et al., 2018) | ✗ | ✗ | ✗ | ✗ | - | 219 | 51 | 0 | 0 | 171,981 |
| C3VG (Yue et al., 2021b) | ✗ | ✗ | ✗ | ✓ | - | 221 | 62 | **5** | 0 | 62,939 |
| SLJA-SYN | ✓ | ✓ | ✓ | ✓ | 4 | 1,237 | 80 | **5** | 136 | 23,913 |
| SLJA-COR | ✓ | ✓ | ✓ | ✓ | 4 | 1,379 | 80 | **5** | 124 | 11,239 |

Table 1: Comparison among the proposed SLJAs dataset and other datasets related to legal judgment. † indicates the datasets are not publically available. SLJA-SYN is a dataset generated using ChatGPT, while SLJA-COR is a dataset derived from SLJA-SYN through manual correction and selection.

| Dataset | CEG | | | | AIG | LJP |
|---|---|---|---|---|---|---|
| | Object | Subject | Objective | Sujective | | |
| SLJA-SYN | 65.7 | 33.3 | 119.7 | 66.4 | 318.5 | 82.7 |
| SLJA-COR | 66.9 | 40.1 | 119.8 | 66.0 | 156.9 | 36.5 |

Table 2: Average length of each subtasks in SLJA-SYN and SLJA-COR.

with bag-of-words. Zhong et al. (2018) combine LSTM models for different LJP task in a topological graph framework. Yang et al. (2019) propose the collocation attention mechanism to improve the performance of LSTM on LJP. Zhou et al. (2022) propose the deep gating network (DGN) to aggregate the features of charges. Zhao et al. (2022) and Lyu et al. (2022) propose reinforcement learning models to extract the sentences of facts that contain criminal elements. Xu et al. (2020) and Zhang et al. (2023) propose to improve the performance of LJP through distinguishing similar charges or articles. Yue et al. (2021a) propose to adopt different parts of facts to conduct judgment predictions. Although these traditional models have exhibited a remarkable proficiency in surpassing the majority baselines legal tasks. The escalation towards more potent LLMs as state-of-the-art (SOTA) benchmarks has become an undeniable trajectory in recent research. More recently, LLMs have been rapidly developed and have become state-of-the-art. Several works start to explore how to design legal prompts to improve the performance of LJP, for instance, Trautmann et al. (2022) introduce the legal prompt engineer for LJP, Yu et al. (2022) utilize the chain-of-thought for legal entailment task, Huang et al.

(2023) release the Lawyer LLaMA, which is fine-tuned with legal datasets (Chen, 2018; Zhong et al., 2020) based on LLaMA model (Touvron et al., 2023). Therefore, we also choose state-of-the-art LLMs as benchmarks in this work. Although these works produce the judgment results according to the facts, they do not provide the analysis process, which makes the judgment results untrustable.

## 2.2 Court View Generation

Court view generation was first proposed to automatically generate rationales for judgment results given facts (Ye et al., 2018). Wu et al. (2020) introduce a counterfactual decoder to eliminate the confounding bias between a fact and its corresponding charges and generate rationales for charges. Li and Zhang (2021) combine two separate optimizers to generate rationales for charges given facts and articles. Yue et al. (2021b) also generate fine-gained rationales for charges and terms of penalty, respectively. Although these works can improve the interpretability of judgment results, they do not demonstrate the logical relationship explicitly in the analysis process.

## 2.3 Legal Judgment Syllogism

Syllogism, as a form of deductive reasoning, is widely used to express logical arguments (Maxeiner, 2011) and determine if a specified behavior is legal or not (Gold, 2018). Syllogism includes three logical components, i.e., major premise, minor premise, and conclusion (d'Almeida, 2019). The major premise comprises the relevant articles, which serve as a theoretical basis. The minor

premise comprises the criminal elements, which provide a factual basis for the reasoning process. The conclusion involves inferring the applicable articles and corresponding judgment results based on the major and minor premises.

In practice, syllogism serves as a standard form of legal reasoning extensively employed by judges to ensure that logical arguments are sound and indisputable (Hart et al., 2012; Constant, 2023). Based on the application of syllogism in real legal scenarios, we hypothesize that completing SLJA would be beneficial for LLMs.

## 3 Task Definition

We define the following four tasks for syllogistic legal judgment analysis (SLJA) based on the concept of syllogistic reasoning:

**Task 1:** Article retrieval (AR) outputs applicable law articles as *major premises* given a fact and article library as inputs.

**Task 2:** Criminal element generation (CEG) outputs four criminal elements as *minor premises* given a fact as an input.

- *Subject* refers to a person with criminal responsibility that committed a crime;
- *Subjective Element* consists of intention and negligence, which refer to The psychological state of the criminal subject towards the criminal behavior and its results;
- *Object* refers to the social relationships protected by criminal law and infringed upon by criminal acts;
- *Objective Elements* refer to the concrete manifestations of crime, which consist of time, location, actions, and consequences.

**Task 3:** Article interpretation generation (AIG) outputs applicable articles and corresponding rationales as part of *conclusion* of a legal judgment given the outputs of AR and CEG as inputs.

**Task 4:** Legal judgment prediction (LJP) outputs the charges and terms of penalty as part of *conclusion* of a legal judgment given the outputs of AIG.

## 4 Dataset Construction

We construct a dataset for legal judgment analysis with syllogistic reasoning.

### 4.1 Raw Data Collection

We obtain criminal cases from the CAIL-Long dataset (Xiao et al., 2021), which are collected from China Judgments Online.[2] We filter charges that occur more than 40 times and randomly select up to 1,000 cases for each charge. Each criminal case contains a fact and its results of the legal judgment, including the charge, term of penalty and related legal articles. In total, we collect 23,913 criminal cases. After obtaining raw data, we first elicit SLJA data using ChatGPT to create SLJA-SYN, and then sample 11,239 cases from SLJA-SYN and manually correct, which is noted as SLJA-COR. The statistics of SLJA datasets are shown in Table 1 and 2. The case distribution of SLJA-COR is shown in Figure 4 (See Appendix B).

### 4.2 Syllogistic Reasoning Data Elicitation

We guide ChatGPT to generate the SLJA dataset by designing specific prompts for each task as inputs (See Appendix A).

The AR task is completed through a two-step iteration: (1) generating the description of articles given a fact. Each fact is used as a query to retrieve the relevant articles from the article library. (2) determining if the retrieved articles can cover the content of the fact. If the response is "No", go back to (1) to conduct retrieval again. Otherwise, the AR task is completed, and the retrieved articles are referred to as the major premise.

For CEG task, we design four separate prompts to: (1) generate the subject, and determine whether the subject should bear criminal responsibility; (2) determine whether the subjective element is intention or negligence, and explain the criminal intent; (3) generate the object according to the fact; (4) generate the objective elements including criminal time, location, actions, and consequences. These generated criminal elements are referred to as the minor premise.

For AIG task, we design a prompt to determine the applicable articles and generate the interpretation according to the outputs of AR and CEG. For LJP task, we design a prompt to generate charges and terms of penalty according to the outputs of AIG. Based on the major premise and minor premise, the outputs of AIG and LJP serve as the conclusion of a legal judgment.

We conduct the following steps to sample a set of ChatGPT generated syllogistic reasoning cases from SLJA-SYN for manual correction: (1) selecting the cases with poor performance of ChatGPT on LJP task; (2) sampling data by charges. We

---

[2] https://wenshu.court.gov.cn/

randomly select up to 500 cases for each charge.

## 4.3 Manual Data Correction

We hire 22 workers with a legal background to work on the correction process. For each task in SLJA, we define three phrase-level operations for correction, including insertion, deletion, and rewriting. We construct an online correction system and provide a guideline for workers. The guideline introduces the definition of each task, the detailed system operations, and an example of data correction. Each worker corrects one case per round, and the system records the corrected positions in the sentences, the corresponding operations, the correction results, and questions during this process. Each worker is responsible for correcting the cases involving 3 to 4 charges.

## 4.4 Data Quality Assurance

To ensure the quality of data correction, we provide: (1) Correction feedback mechanism. We collect questions during the correction process, consult with experts for solutions, and share the question-solution pairs with all workers. (2) Automatic quality estimation. For each corrected data, we compare the corrected charges with ground-truth charges, if the corrected charges do not contain the ground-truth charges, then we request another worker to correct it again. (3) Manual sampling inspection. For each charge, we randomly sample 10% data and request another worker to check the miss-corrected and grammatical issues. If there are errors in the corrected data, we require the original worker to re-correct the data.

## 5 Experiments

### 5.1 Benchmark Models

We employ four LLMs as decoders to generate task outputs, which serve as benchmark models with strong performance in Chinese.

- **ChatGLM** (Du et al., 2022) adopts the traditional feed-forward network with model parameters of 6.2 billion. It conducts supervised fine-tuning and reinforcement learning from human feedback (RLHF) training on a mixed Chinese and English dataset.
- **Davinci002**[1] adopts transformer-based structure with model parameters of 175 billion, conducts supervised instruction tuning on a multilingual corpus (Fu et al., 2022).

- **Davinci003**[1] is trained with RLHF method based on the Davinci002. Davinci003 improves the capability of in-context learning (Fu et al., 2022).
- **ChatGPT**[1] is also trained with RLHF method based on the Davinci002. ChatGPT improves the capabilities of in-context learning and dialogue context modeling (Fu et al., 2022).

We also evaluated ChatYuan (Xuanwei Zhang and Zhao, 2022), BELLE-GLM (Yunjie Ji and Li, 2023), and Vicuna (Chiang et al., 2023), but they cannot understand the concepts of four criminal elements, so they cannot follow instructions to conduct syllogistic reasoning.

In the experiments, we use the prompt templates as shown in Table 6 (See Appendix A), and to ensure the reproducibility, we set the hyper-parameter *temperature=0* to control the randomness of generation.

### 5.2 Evaluation Metrics

We use R@k for applicable article prediction and charge prediction, instead of classification metrics (Gan et al., 2022), the reason is that the LLMs produce more than one possible correct answer for each task.

We add a penalty factor in **R@k** metric for term of penalty prediction as shown in Eq. 1,

$$R_t@k = \frac{\sum_i^N \mathbb{1}[t \in \tilde{T}_i[:k] \cap T_i] \cdot a_i}{\sum_i^N \mathbb{1}[t \in T_i]},$$

$$a_i = \begin{cases} \frac{1}{max(1,log_{12}(l_i))} & t \in \text{imprisonment} \\ 1 & \text{otherwise} \end{cases},$$

where $a_i$ is the penalty factor, which aims to penalize the ambiguous range of predicted imprisonment with a lower recall score. The greater the ambiguity of the predicted imprisonment, the lower the penalty factor, resulting in a decreased recall score. $t$ indicates the type of term of penalty, $N$ indicates the data size, $l_i$ indicates the scope of imprisonment, $T_i$ indicates the ground-truth and $\tilde{T}_i[:k]$ indicates top-k results in predicted terms of penalty. When the scope is less than 12 months, we do not punish the results, otherwise, the penalty factor decreases as the scope increases.

For CEG task, we extract criminal elements from the generated results by regular expressions and evaluate these elements with specific metrics, respectively. Precision (**P**), Recall (**R**), and **F1** score

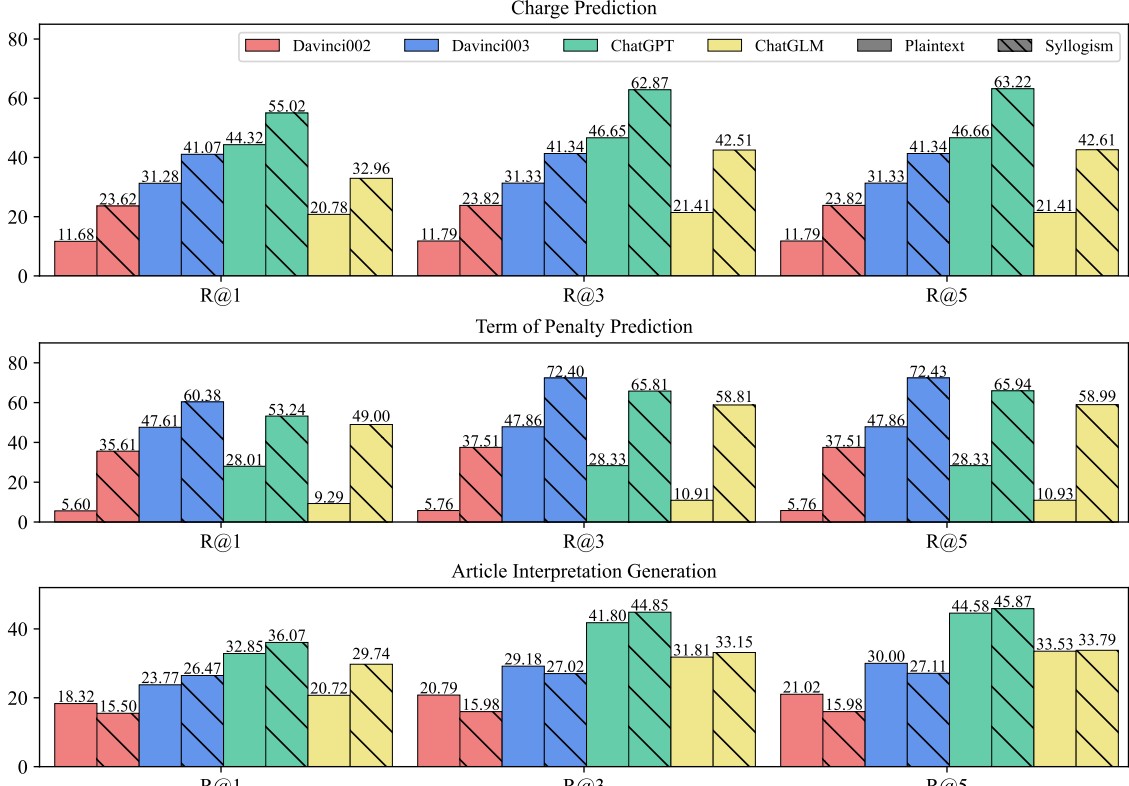

Figure 2: Main performance, including the results of charge prediction, term of penalty prediction and article interpretation generation.

| Model | Subjective Element | | | Subject | | Objective Elements | Object | |
| | P (%) | R (%) | F (%) | Match (%) | ROUGE (%) | ROUGE (%) | Match (%) | ROUGE (%) |
|---|---|---|---|---|---|---|---|---|
| ChatGLM | 76.96 | 53.32 | 59.95 | 79.41 | 39.90 | 52.51 | 16.56 | 27.15 |
| Davinci002 | **94.96** | 51.64 | 51.60 | 94.18 | 33.92 | 61.51 | 9.40 | 23.71 |
| Davinci003 | 92.22 | 69.25 | 75.95 | 93.07 | 48.45 | 66.32 | 17.85 | 34.23 |
| ChatGPT | 82.25 | **95.66** | **89.60** | **99.61** | **77.76** | **96.04** | **48.26** | **86.41** |

Table 3: The performance of criminal element generation.

are used to evaluate the performance of subjective element prediction, which is a binary classification task to determine if intention or negligence. **Match** score evaluates the exact matches between generated items and human-corrected ones. **ROUGE** score evaluates the word overlap between generated subject, objective elements, and object and manually corrected ones.

### 5.3 Main Performance

We present the judgment performance of all the LLMs, both with syllogism and without syllogism (i.e., plaintext) on the SLJA-SYN dataset. (See Figure 2).

First, for charge prediction and term of penalty prediction, all models perform better with syllogism than without syllogism. For example in the charge prediction, ChatGPT with syllogism

reaches 55.02/62.87/63.22 on R@1/R@3/R@5, which are 10.70/16.22/16.56 superior then plaintext, and in the term of penalty prediction, ChatGPT with syllogism reaches 53.24/65.81/65.94 on R@1/R@3/R@5, which are 25.23/37.48/37.61 superior then plaintext. This indicates that the tasks of AR, CEG and AIG are of great benefit for the LJP task.

Second, for charge prediction, ChatGPT outperforms Davinci003 model. This is because the models generate charges based on applicable articles and Davinci003 model performs worse on the AIG task. As we can see in the Article Interpretation Generation, ChatGPT reaches 36.07/44.85/45.87 on R@1/R@3/R@5, which is 9.60/17.83/18.76 points higher than Davinci003.

Thirdly, for penalty term prediction, both

| Setting | Charge | | | Article | | | Term of Penalty | | |
|---|---|---|---|---|---|---|---|---|---|
| | R@1 | R@3 | R@5 | R@1 | R@3 | R@5 | R@1 | R@3 | R@5 |
| ChatGPT | **55.02** | **62.87** | **63.22** | **36.07** | **44.85** | **45.87** | 53.24 | 65.81 | 65.94 |
| -object | 54.27 | 62.31 | 63.00 | 35.88 | 44.10 | 45.27 | 49.89 | 59.68 | 60.08 |
| -objective | 49.98 | 58.65 | 59.06 | 33.33 | 42.81 | 43.58 | 43.94 | 60.58 | 60.86 |
| -subject | 54.91 | 62.62 | 62.92 | 35.96 | 44.04 | 45.14 | 52.42 | 64.54 | 64.72 |
| -subjective | 54.30 | 61.91 | 62.31 | 35.62 | 43.72 | 44.66 | **53.32** | **65.93** | **66.03** |
| -criminal elements | 52.30 | 60.00 | 60.63 | 35.26 | 42.25 | 43.83 | 46.20 | 54.26 | 54.73 |

Table 4: The ablation study of criminal elements, including removing one criminal element and replacing all criminal elements with fact (denoted as -criminal elements). Bold and underlined fonts indicate leading and compared results in each setting.

| #Iteration | Charge | | | Article | | | Term of Penalty | | |
|---|---|---|---|---|---|---|---|---|---|
| | R@1 | R@3 | R@5 | R@1 | R@3 | R@5 | R@1 | R@3 | R@5 |
| 3 | 55.02 | 62.87 | 63.22 | 36.07 | 44.85 | 45.87 | 53.24 | 65.81 | 65.94 |
| 2 | 54.24 | 63.06 | 63.78 | 35.50 | 43.61 | 45.30 | 50.52 | 64.06 | 64.36 |
| 1 | 52.04 | 58.17 | 58.44 | 33.19 | 39.31 | 40.34 | 49.09 | 61.74 | 61.83 |
| 0 | 39.86 | 41.55 | 41.57 | - | - | - | 25.13 | 37.86 | 37.86 |

Table 5: The ablation study of maximum iteration number in article retrieval. 0 indicates removing the AR task, and only uses minor premises as the input of LJP.

ChatGLM and Davinci002 demonstrate significantly better performance when utilizing syllogism compared to plaintext. As we can see in the term of penalty prediction, Davinci002 and ChatGLM with syllogism reach 35.61/37.51/37.51 and 49.00/58.81/58.99 respectively on R@1/R@3/R@5, which are 30.01/31.75/31.75 and 39.71/47.90/48.06 superior to plaintext. The reason is that the Davinci002 and ChatGLM without syllogism always output meaningless output, such as "there is not enough information to determine the sentence". This indicates that the syllogism method can provide enough useful information to help the model generate meaningful terms of penalty.

Fourth, in the prediction of terms of penalty, Davinci003 outperforms ChatGPT in all metrics. As we can see in the Term of Penalty, Davinci003 reaches 60.38/72.40/72.43, on R@1/R@3/R@5, which outperforms ChatGPT by 7.14/6.59/6.49. The reason is that ChatGPT tends to predict more middle or severe terms of penalty, the distribution is shown in Figure 3.

Fifth, for article interpretation generation, Davinci002 with syllogism performs worth than plaintext on all metrics, while Davinci003 with syllogism outperforms plaintext on R@1 and performs close on other metrics. This indicates that

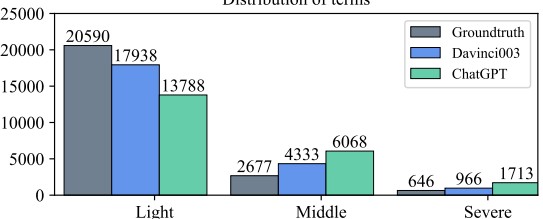

Figure 3: The distribution of Terms of Penalty. Light refers to the light penalty, including imprisonment with less than three years, detention and surveillance. Mid refers to the middle penalty, including imprisonment with more than three years and less than ten years. Severe refers to the severe penalty, including imprisonment with more than ten years, life penalty and death penalty.

RLHF training method is beneficial for the syllogism method. The reason may be that syllogism is more close to the human thinking mode.

## 5.4 Criminal Element Generation Analysis

We report the results of CEG task on corrected SLJA-COR dataset in Table 3.

First, ChatGPT outperforms all the other models on all metrics except P in the subjective element. Specifically, objective elements and the subject are contained in the facts which can be extracted by LLMs, and the subjective element and the object are not contained in the facts which need to be inferred from the facts. The reason is that Chat-

GPT has stronger capabilities of understanding and deduction of fact description.

Second, for the subject, Davinci003 performs worse than Davinci002 on Match metric. Specifically, Davinci003 is 14.53 superior on ROUGE metric but 1.11 inferior on Match metric. This indicates that both the Davinci002 and Davinci003 can accurately extract the subject, but the analysis part of Davinci003 is closer to the manual corrected one. This reason may be the RLHF training in Davinci003 can align the model with humans.

Third, for the object, other models outperform Davinci002 on Match and ROUGE metrics. As we can see in the object of Table 3, Davinci002 is 7.16, 8.45 and 38.86 inferior on Match metrics than ChatGLM, Davinci003 and ChatGPT respectively. The reason is that Davinci002 cannot understand the concept of the object, it refers to the object as somebody or something that has been violated by a criminal, rather than a social relationship.

### 5.5 Effects of Criminal Elements

In Table 4, we report the ablation study of criminal elements.

First, the objective is much helpful for legal judgment. As we can see in Table 4, removing the objective elements makes the performance of R@1/R@3/R@5 drops 5.04/4.22/4.16, 2.74/2.04/2.29 and 9.30/5.23/5.08 in the prediction of charge, article, and term of penalty respectively. The reason is that the objective elements describe the characteristics of the crime (time, location, and behavior) and the consequences of the crime, which can be referred to as the summary of a fact.

Second, using fact as the minor premise is inferior on all metrics to criminal elements. The performances of -criminal elements on R@1/R@3/R@5 are 2.72/2.87/2.59, 0.81/2.60/2.04, and 7.04/11.55/11.21 inferior in the prediction of charge, article and the terms of a penalty than full prompt respectively. The reason is that the four criminal elements can provide more comprehensive information for the model than only using facts.

Third, the object has a significant impact on the term of penalty prediction. Specifically in Table 4, the performance without the object (-object) is 3.35/6.13/5.86 inferior to the full prompt. The reason is that the object can determine the severity of the crime, leading to different terms of penalty.

Fourth, subject and subjective elements have lit-

tle impact on decision prediction. The reason is that in most cases, a subject is a natural person with full capacity for the conduct, while in reality, the subjective element of most cases are intentional, so these two elements can only bring a minor improvement.

### 5.6 Ablation of Major Premises

In Table 5, we report the ablation study of the maximum number of article retrieval.

First, As the maximum number of searches increases, the performance of all metrics continuously improves except R@3 and R@5 in charge prediction. The reason is that during each iteration, ChatGPT will determine whether the currently retrieved methods meet the requirements, so it is possible to supplement the correct articles that were not retrieved before. This indicates that repeatedly asking ChatGPT can correct the retrieved articles.

Second, without article retrieval, the performance of charge and term of penalty prediction has a significant decrease. This indicates the criminal elements need to be combined with retrieved articles can obtain reasonable judgment results.

Third, for the prediction of charge and article, the performance of maximum iteration number of 3 is close to 2, as we can see in Table 5, the improvements of R@1/R@3/R@5 on the prediction of charge and article are 0.78/-0.19/-0.56 and 0.57/1.24/0.57 respectively. This indicates that ChatGPT conducts only one check can achieve good results.

## 6 Conclusion and Future Work

In this paper, we introduce the resources of syllogistic reasoning for legal judgment analysis: the task, dataset and benchmarks. We define the SLJA according to the concept of syllogistic reasoning, including four tasks: article retrieval (AR), criminal element generation (CEG), article interpretation generation (AIG) and legal judgment prediction (LJP). We release the first Chinese SLJA dataset. The dataset is constructed by correcting the outputs of each task, which are generated by ChatGPT. In order to ensure the quality of the dataset, we hire workers with a legal background for data correction. Each case in SLJA dataset includes a detailed analysis process of syllogistic reasoning and we expect the dataset will facilitate the development of trustworthy legal judgment assistants. Besides, we also collect four state-of-the-art LLMs for

extensive experiments. We design specific prompts for each task as inputs of LLMs. We evaluate each model with and without syllogism and conduct an in-depth analysis. The experimental results indicate that: (1) SLJA is still challenging for LLMs. (2) Syllogistic reasoning can provide effective information for LLMs to improve LJP. (3) RLHF training is beneficial for syllogistic reasoning. (4) The major premise and minor premise in syllogism are both crucial for the final judgment results which can provide more detailed interpretation to make the legal judgment assistant trustable.

As to future work, on the one hand, we will extend SLJA dataset with more legal types and theories, such as civil law and two-stage theory. On the other hand, this work is currently releasing a test set for LLM evaluation. In the future, we will construct more data to train and evaluate traditional models and conduct a human study to enhance the practicality and interpretability of our approach. Legal LLMs (e.g., Lawyer LLaMA) that have emerged during the same period as this work will be evaluated in future research. Last but not least, we call for studies to improve the benchmark performance, as well as conduct underexplored research, such as legal judgment analysis for rare charges under low-resource settings.

## 7 Reproducibility

To promote the development of legal analysis tasks and facilitate the reproducibility of the results reported in this paper, our released datasets and codes are available at https://github.com/dengwentao99/SLJA.

## Limitations

Note that while syllogistic reasoning serves as a guide for analyzing legal arguments, it is not rigidly applied to overturn legal decisions in practice. Judges also take into consideration other factors, such as statutory law, case law, public policy, and the specific facts of the case. Instead, it is a useful and practical tool to assist judges in analyzing legal arguments and assessing whether the law has been correctly applied to the parties' arguments. This approach helps maintain consistency and fairness in the decision-making process. Additionally, our proposed dataset will be extended to evaluate traditional models. Evaluation for the LLMs fine-tuned by legal datasets that have emerged during the same period as this work will be conducted in the future.

## Ethics Statements

We acknowledge the potential risks in the research of legal judgment assistants, and thus, it is imperative to address ethical issues in the legal decision-making process. It is crucial to predict legal judgment results while also taking into account ethical concerns such as safeguarding personal privacy and respecting the authority of the law articles. All raw data collected for this study are sourced from the publicly accessible website and have undergone ethical considerations by the website administrators and dataset publishers, in which a majority of cases have undergone anonymization to safeguard sensitive information.

## Acknowledgement

We would like to thank the editors and reviewers for their helpful comments. This research was supported by the National Key R&D Program of China (No.2022YFC3303004, No.2020YFB1406704), the Natural Science Foundation of China (62102234, 62272274, 62202271, 61902219, 61972234, 62072279), the Key Scientific and Technological Innovation Program of Shandong Province (2019JZZY010129), the Fundamental Research Funds of Shandong University, and VOXReality (European Union Grant, No. 101070521). All content represents the opinion of the authors, which is not necessarily shared or endorsed by their respective employers and/or sponsors.

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

## A   Prompt templates

The Prompt templates are shown in Table 6.

## B   Case distribution

Figure 4 demonstrates the distribution of cases, which shows the charges with the number of cases exceeding 50.

| Task | | Prompt template |
|------|------|-----------------|
| AR | Generation | 请根据基本案情列举出相关的法条。
Please list the relevant articles based on the fact. |
| | Determination | 请判断当前检索得到的法条是否能够概括基本案情中的所有内容，
请回答是或者否并说明原因。
Please determine if the retrieved articles can cover the content of the fact,
Please answer "Yes" or "No" and explain it. |
| CEG | Subject | 请根据基本案情，抽取犯罪主体，并分析犯罪主体的构成要件。
犯罪主体的构成要件包括：是否达到法定年龄、是否是完全行为能力人。
Please extract subject from the fact and analyze the constituent elements of subject.
Constituent elements of subject includes reaching criminal age and being a fully
capable person. |
| | Subjective Element | 请根据基本案情，抽取犯罪主观方面，并说明犯罪意图。
犯罪主观方面包括：故意和过失。
Please extract subjective element from the fact and describe the criminal intents.
The subjective element includes: intention and negligence. |
| | Object | 请根据基本案情，抽取犯罪客体，并给出犯罪客体的分析。
犯罪客体：刑法所保护而为犯罪所侵犯的社会关系。
Please extract the object from the fact and explain it.
Object: The social relationship protected by criminal law and infringed upon by the crime. |
| | Objective Elements | 请根据基本案情，抽取犯罪的客观方面。
犯罪客观方面包括：犯罪时间、犯罪地点、犯罪行为、犯罪结果。
Please extract the objective elements from the fact.
Objective elements include: criminal time, location, actions and consequences. |
| AIG | | 请根据犯罪要件从相关法条中匹配出符合的法条，
列举适用的法条并说明每一个法条匹配的原因。
Please determine the applicable articles from relevant articles according to criminal elements.
List the applicable articles and explain them. |
| LJP | | 请根据适用法条的分析结果，给出判决结果，判决结果包括：1.罪名；2.刑期。
Please generate the judgment results according to the applicable articles,
the judgment results include: 1. charges; 2. terms of penalty. |

Table 6: The prompt templates for each task.

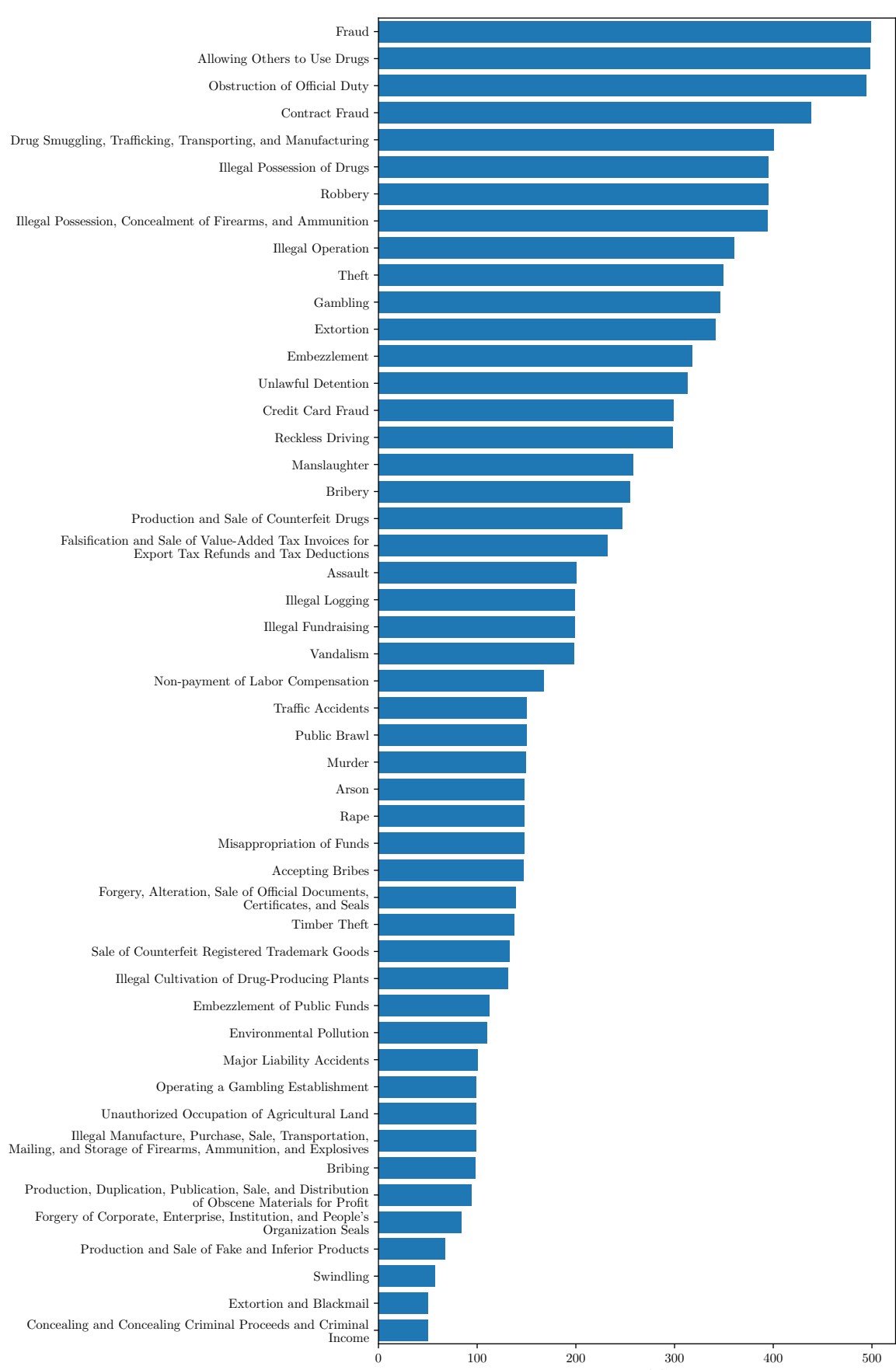

Figure 4: The distribution of cases in SLJA-COR dataset, in which the number of cases for each charge is larger than 50.