# OpenReview forum: "Syllogistic Reasoning for Legal Judgment Analysis"
_EMNLP/2023/Conference — EMNLP 2023 Main_

### Official Review · Reviewer_vyvp · 2023-07-30

**Soundness:** 3

**Excitement:**

3: Ambivalent: It has merits (e.g., it reports state-of-the-art results, the idea is nice), but there are key weaknesses (e.g., it describes incremental work), and it can significantly benefit from another round of revision. However, I won't object to accepting it if my co-reviewers champion it.

**Paper Topic And Main Contributions:**

In this paper, authors construct a syllogistic reasoning dataset for legal judgment analysis. Besides the authors compare a lot of LLMs, conducing an in-depth analysis of the capacity of their legal judgement analysis.

**Reasons To Accept:**

1.It is necessary to use LLMs for legal judgement analysis.

2.Authors design a Syllogistic Reasoning approach for Legal Judgment Analysis and demonstrate the effectiveness of multiple LLMs on legal judgment tasks.

**Reasons To Reject:**

1.Whether the authors have tried to experiment with LLMs that have been fine-tuned using legal data, e.g., Lawyer LLaMA [1].

2.Authors propose a four-criminal element approach to analyse legal judgment is feasible, but at present, the four-element system receives a lot of controversy, whereas the two-stage system is currently the dominant theory for analysing legal judgment, which the authors could explore further [2].

3.The authors can compare some classical methods to show the improvement of LLMs.

4.The authors can discuss the feasibility in civil law, not only limited to criminal law.

5.More cases are needed.


References

[1] Quzhe Huang et al. 2023. Lawyer LLaMA Technical Report

[2] Mingkai Zhang et al. 2003. Criminal Law


**Reproducibility:**

4: Could mostly reproduce the results, but there may be some variation because of sample variance or minor variations in their interpretation of the protocol or method.

**Reviewer Confidence:**

5: Positive that my evaluation is correct. I read the paper very carefully and I am very familiar with related work.

---

> ### Author Rebuttal · Authors · 2023-08-29
>
> **Comment 1:**
>
> Whether the authors have tried to experiment with LLMs that have been fine-tuned using legal data, e.g., Lawyer LLaMA [1].
>
> **Response:**
>
> **Our work and Lawyer Llama emerged during the same period**, both our submission date and the release date of the Lawyer Llama model fall in the same month of June. Your suggestion holds significant value, and we greatly appreciate it. In forthcoming endeavors, we intend to include legal fine-tuning models such as Lawyer LLaMA for comparison, enhancing the comprehensiveness of our research.
>
> &nbsp;
>
> **Comment 2:**
>
> Authors propose a four-criminal element approach to analyse legal judgment is feasible, but at present, the four-element system receives a lot of controversy, whereas the two-stage system is currently the dominant theory for analysing legal judgment, which the authors could explore further [2].
>
> References
>
> [1] Quzhe Huang et al. 2023. Lawyer LLaMA Technical Report
>
> [2] Mingkai Zhang et al. 2003. Criminal Law
>
> **Response:**
>
> **In the Chinese legal system, the Four Elements theory is commonly employed as a theoretical foundation.**
>
> The Four Elements theory features a rigorous logic and aligns with cognitive principles. It can distinctly delineate the boundaries between guilt and innocence. Therefore, compared to the Two-stage system Theory, the Four Elements theory is more in line with China's requirements for legal analysis [1-3].
>
> &nbsp;
>
> **Comment 3:**
>
> The authors can compare some classical methods to show the improvement of LLMs.
>
> **Response:**
>
> **Our goal is to comprehensively assess the influence of syllogism on State-of-the-Art (SOTA) LLMs for legal tasks, instead of outperforming classical methods.**
>
> Although classical BERT-based models [1-3] have exhibited a remarkable proficiency in surpassing majority baselines when tackling legal tasks within the scope of legal tasks. This pronounced advantage primarily stems from their robust language modeling capabilities. The escalation towards more potent Large Language Models (LLMs) as SOTA benchmarks has become an undeniable trajectory in recent research. In alignment with this trajectory, we have meticulously assembled an exhaustive compilation of cutting-edge LLMs, drawing from our best available knowledge. These curated models will serve as the cornerstone for our benchmarking standards. We focus on investigating the influence of syllogism on LLMs and this approach opens pathways for integrating advanced models into the framework instead of comparing them to classical methods.
>
> &nbsp;
>
> **Comment 4:**
>
> The authors can discuss the feasibility in civil law, not only limited to criminal law.
>
> **Response:**
>
> **Our proposed framework is firmly grounded in practical utility, while we recognize the value of discussing its feasibility in civil law contexts.**
>
> It remains pertinent to acknowledge that the theory of elements of a crime, rooted in the evolution of criminal law [1], holds no applicability within the domain of civil law. While our framework's context primarily engages with criminal elements as minor premises, acknowledging and comparing these distinctions remains constructive. Adding such a comparative discussion below ensures our work's accessibility across domains, fostering comprehensive understanding among readers from diverse backgrounds.
>
> Civil law and criminal law exhibit substantial disparities in their initiation, resolution processes, and overarching objectives. In criminal law, cases address offenses against society, resulting in punitive measures against wrongdoers [4]. Conversely, civil law governs disputes between individuals or entities, emphasizing restitution for aggrieved parties [5]. Criminal cases may entail penalties like imprisonment, while civil cases aim to redress losses [6,7]. This stark contrast underscores the profound divergence in purpose and consequences between the two legal realms.
>
> &nbsp;
>
> **Comment 5:**
>
> More cases are needed.
>
> **Response:**
>
> As shown in Table 1, our proposed benchmark dataset encompasses the largest count of cases involving complete subtasks of syllogistic reasoning. Furthermore, our approach to data annotation is adaptable, allowing for potential expansion of the dataset in the future.
>
> &nbsp;
>
> **References:**
>
> [1] Mingxuan Gao. 2009. Criminal Law.
>
> [2] Zhao, J., et al. 2022. Charge prediction by constitutive elements matching of crimes. IJCAI.
>
> [3] Yougang Lyu et al. 2022. Improving legal judgment prediction through reinforced criminal element extraction. IPM.
>
> [4] P.J.A. Von Feuerbach. 1847. Lehrbuch des gemeinen in Deutschland gültigen peinlichen Rechts. Georg Friedrich Heyer.
>
> [5] Yuan Zhi-jie. 2020. The Civil Basis of State liability. Journal of Political Science and Law.
>
> [6] Civil Code of the People's Republic of China. 2020.
>
> [7] Criminal Procedure Law of the People's Republic of China. 2018.

---

### Official Review · Reviewer_mGTi · 2023-08-03

**Soundness:** 3

**Excitement:**

2: Mediocre: This paper makes marginal contributions (vs non-contemporaneous work), so I would rather not see it in the conference.

**Paper Topic And Main Contributions:**

This paper mainly constructs a new dataset called SLJA, according to the concept of syllogistic reasoning, including four tasks: article retrieval (AR), criminal element generation (CEG), article interpretation generation (AIG) and legal judgment prediction (LJP). Besides, the authors evaluate the performance of four popular LLMs on the SLJA dataset.

**Questions For The Authors:**

Question A. Why use LLM to do the prediction?

Question B. The abstract say there are 11,239 cases, is it all processed by the Manual Data Correction?


**Reasons To Accept:**

The dateset is valuable with the human correction.

This writing is clear and the paper is easy to follow.


**Reasons To Reject:**

Lack of experiments. The authors only use LLMs in the experiments, and such experiment settings are not sufficient enough to prove the value of the dataset, since the new content is only used as in-context learning but not in the model training.

Besides, why use LLM to do the prediction? Especially in LJP task, there are so many labels to predict, and it’s difficult for LLMs. Moreover, the inference speed of LLMs are much slower than the small predictive models, and the prediction results of LLMs are also un-controlable (one irrelevant word can change the whole result). In fact, the sota LJP models already achieve much better performance (e.g., LADAN[1] or NeurJudge[2]).

Some details of dataset are not clear, at least there should be a data analysis, including the distribution of the cases, the average length of each tasks and so on.

[1] Distinguish Confusing Law Articles for Legal Judgment Prediction

[2] NeurJudge: A Circumstance-aware Neural Framework for Legal Judgment Prediction


**Reproducibility:**

3: Could reproduce the results with some difficulty. The settings of parameters are underspecified or subjectively determined; the training/evaluation data are not widely available.

**Reviewer Confidence:**

4: Quite sure. I tried to check the important points carefully. It's unlikely, though conceivable, that I missed something that should affect my ratings.

---

> ### Author Rebuttal · Authors · 2023-08-29
>
> **Comment 1:**
>
> Lack of experiments. The authors only use LLMs in the experiments, and such experiment settings are not sufficient enough to prove the value of the dataset, since the new content is only used as in-context learning but not in the model training.
>
> **Response:**
>
> **Our experiments focus on assessing the influence of syllogism on State-of-the-Art (SOTA) LLMs for legal judgment analysis.**
>
> We aim to provide a benchmark dataset for assessing the influence of syllogism on LLMs.
> The syllogism reasoning legal judgment analysis benchmark dataset contains all subtasks in syllogism, syllogism is a commonly used method for reasoning in legal judgment to analyze cases and arrive at judgment results in real practice[1,2].  Although BERT-based models excel in legal tasks [1-3], legal research on Large Language Models (LLMs) is a frontier trend [1], which is the basis of forming legal task benchmarks.  In the experiments, we employ the prompt learning approach which is suitable for inference in large models [4,5], instead of the training methods used in the traditional LJP models.
>
> &nbsp;
>
> **Comment 2:**
>
> Besides, why use LLM to do the prediction? Especially in LJP task, there are so many labels to predict, and it’s difficult for LLMs. Moreover, the inference speed of LLMs are much slower than the small predictive models, and the prediction results of LLMs are also un-controlable (one irrelevant word can change the whole result). In fact, the sota LJP models already achieve much better performance (e.g., LADAN[1] or NeurJudge[2]).
>
> [1] Distinguish Confusing Law Articles for Legal Judgment Prediction
>
> [2] NeurJudge: A Circumstance-aware Neural Framework for Legal Judgment Prediction
>
> **Response:**
>
> **Compared to the traditional LJP model, LLMs can offer a more extensive reasoning process to assist judges in judgment decisions.**
>
> Although traditional BERT-based models [1-3] have exhibited a remarkable proficiency in surpassing majority baselines when tackling legal tasks within the scope of legal tasks. This pronounced advantage primarily stems from their robust language modeling capabilities. The escalation towards more potent Large Language Models (LLMs) as SOTA benchmarks have become an undeniable trajectory in recent research. In alignment with this trajectory, we have meticulously assembled an exhaustive compilation of cutting-edge LLMs, drawing from our best available knowledge. These curated models will serve as the cornerstone for our benchmarking standards. This strategic approach effectively eliminates the need for relying on conventional majority baselines, paving the way for the integration of advanced models into the framework.
>
> Although generating complete reasoning steps takes more time compared to predicting a single judgment result, it can offer more explanations to foster judges' trust in the model.
>
> Some previous works have demonstrated that varying the prompts with minor irrelevant content does not alter the output results[4].
>
> &nbsp;
>
> **Comment 3:**
>
> Some details of dataset are not clear, at least there should be a data analysis, including the distribution of the cases, the average length of each tasks and so on.
>
> **Response:**
>
> Providing a comprehensive dataset analysis is indeed valuable, and we acknowledge your point. We appreciate your suggestion, and we have incorporated the Case distribution of the top-10 charges in Table 1. Additionally, we've included the average length of each task in Table 2 to enhance the clarity of our dataset's characteristics. We believe the supplementary information we have added below contributes positively to our work.
>
> **Table 1.** Case distribution of top-10 charges.
> |    &nbsp;  |    &nbsp;   |    &nbsp;    |    &nbsp;    |    &nbsp;    |    &nbsp;   |
> |    :-:   |    :-:   |    :-:    |    :-:    |    :-:    |    :-:    |
> |    **Charge**      | Fraud |  Harboring Drug Users    | Obstruction of Official Duties | Contract Fraud | Smuggling, Trafficking, Transporting, and Manufacturing Drugs |
> |   **Proportion**  |    4.85%   |    4.84%    |   4.40%    |    3.91%    |   3.56%    |
> | **Charge**      | Illegal Possession of Drugs | Robbery | IIllegal Possession and Concealment of Firearms and Ammunition | Illegal Business Operation | Theft |
> |   **Proportion**   |    3.52%  |   3.52%    |    3.51%   |   3.21%    |    3.11%   |
>
>  &nbsp;
>  &nbsp;
>  &nbsp;
>
> **Table 2.** The average length of each task.
>
> |    &nbsp;  |    &nbsp;   |    &nbsp;    |    &nbsp;    |    &nbsp;    |    &nbsp;   |    &nbsp;   |    &nbsp;   |
> |    :----:   |    :----:   |    :----:    |    :----:    |    :----:    |    :----:    |    :----:    |    :----:    |
> | **Subtask** |     AR    | Object | Objective | Subject | Subjective |  AIG |   LJP  |
> | **Avg. #Task**  |    1217.3   | 66.8 | 119.8 | 40.2 | 65.9 |  121.3 |   36.5 |
>
> &nbsp;
>
> **Question A. Why use LLM to do the prediction?**
>
> **Response:**
>
> Although traditional BERT-based models [1-3] have exhibited a remarkable proficiency in surpassing majority baselines when tackling legal tasks within the scope of legal tasks. This pronounced advantage primarily stems from their robust language modeling capabilities. The escalation towards more potent Large Language Models (LLMs) as SOTA benchmarks have become an undeniable trajectory in recent research. In alignment with this trajectory, we have meticulously assembled an exhaustive compilation of cutting-edge LLMs, drawing from our best available knowledge. These curated models will serve as the cornerstone for our benchmarking standards.
>
> &nbsp;
>
> **Question B. The abstract say there are 11,239 cases, is it all processed by the Manual Data Correction?**
>
> **Response:**
>
> Yes, you are right, all 11,239 cases are processed by manual data correction. We hired 22 workers with a legal background to work on the correction process.
>
> &nbsp;
>
> **References:**
>
> [1] Cui Junyun et al. 2022. A survey on legal judgment prediction: Datasets, metrics, models and challenges. Arxiv.
>
> [2] Gil Semo et al. 2022. ClassActionPrediction: A Challenging Benchmark for Legal Judgment Prediction of Class Action Cases in the US. ACL.
>
> [3] Ashutosh Modi. 2023. SemEval 2023 Task 6: LegalEval - Understanding Legal Texts. ACL.
>
> [4] Trautmann et al. 2022. Legal prompt engineering for multilingual legal judgment prediction. Arxiv.
>
> [5] Blair-Stanek et al. 2023. Can GPT-3 perform statutory reasoning?. Arxiv.

---

### Official Review · Reviewer_JXs1 · 2023-08-04

**Soundness:** 5

**Excitement:**

4: Strong: This paper deepens the understanding of some phenomenon or lowers the barriers to an existing research direction.

**Paper Topic And Main Contributions:**

This paper breaks down Legal Judgment Prediction as a sequence of tasks. The point is to introduce interpretability and explainability. The paper describes how a set of cases from Chinese courts were annotated for each task. The annotation was done as a prediction made by ChatGPT followed by post-editing by crowdworkers. Finally, the paper compares various LLMs on this new dataset.

**Questions For The Authors:**

A. Why add a penalty factor in Eq 1 on Line 360? What is the meaning of that factor? I.e. what does the penalty stand for, what behavior of the model does it penalize?

B. Figure 2: there is no baseline or previous SOTA to compare to. What score would a majrity baseline get?

C. The ethics statement says that sensitive information has been anonymized by the authors (Line 590). How was that done?

D. How were the LLMs prompted? Was there any randomness in the generation? This is important for reproducibility.

**Reasons To Accept:**

- This paper presents a principled way of introducing interpretability into legal AI decisions.
- This paper introduces a benchmark dataset, and the annotation process is sound.

**Reasons To Reject:**

- It is not clear whether the proposed framework is actually interpretable for humans; it's possible that output produced by models following this framework are not useful in practice. A human study could help answer that question.

**Reproducibility:**

4: Could mostly reproduce the results, but there may be some variation because of sample variance or minor variations in their interpretation of the protocol or method.

**Reviewer Confidence:**

4: Quite sure. I tried to check the important points carefully. It's unlikely, though conceivable, that I missed something that should affect my ratings.

**Typos Grammar Style And Presentation Improvements:**

- The colors in Figures 2 and 3 could be more color-blind friendly. See https://www.extensis.com/hs-fs/hubfs/Colorblind-Color-Palette.jpg?width=15003&name=Colorblind-Color-Palette.jpg for ideas.
- The related work around Lines 147-163 enumerates papers and summarizes them. It would be more helpful to provide an overview, e.g. saying that many papers have addressed the task of LJP, and then cite a string of papers without going into too much detail.
- Line 363 "scope of imprisonment" Do you mean "duration of imprisonment"?
- "plaintext": this word is used multiple times, for example on Line 390. From the paper, I'm guessing that it means "without syllogism", i.e. prompting a model to predict charges without prompting it for the intermediate tasks. To help the reader understand, it would be good to clarify this, for example with a footnote.

---

> ### Author Rebuttal · Authors · 2023-08-29
>
> **Comment:**
>
> It is not clear whether the proposed framework is actually interpretable for humans; it's possible that output produced by models following this framework are not useful in practice. A human study could help answer that question.
>
> **Response:**
>
> **Our proposed framework is designed to offer actual interpretability and is rooted in practical utility.**
>
> We hired 22 workers with a legal background to build the dataset. We asked their advice and confirmed that, in the real judgment practice, syllogism is a commonly used method of reasoning in legal judgment analysis to analyze cases and arrive at judgment results [1-3]. This serves as a strong basis for the interpretability of our framework in a real-world legal context.
>
> Furthermore, past research has demonstrated that the reasoning framework applied to LLM can emulate human-like reasoning processes, lending credence to its practical relevance [3].
>
> While we acknowledge your concern about the practical utility of our framework, we are confident that it can yield valuable insights and meaningful results. We appreciate your suggestion of incorporating a human study to further demonstrate the effectiveness of our proposed framework.  In our future work, we will certainly explore the possibility of conducting a human study to enhance the practicality and interpretability of our approach. Thank you for your constructive comments, which will undoubtedly contribute to the improvement of our work.
>
> &nbsp;
>
>
> **Question A:**
>
> Why add a penalty factor in Eq 1 on Line 360? What is the meaning of that factor? I.e. what does the penalty stand for, what behavior of the model does it penalize?
>
> **Response:**
>
> **The penalty factor aims to penalize the ambiguous range of predicted imprisonment with a lower recall score.**
>
> The greater the ambiguity of the predicted imprisonment, the lower the penalty factor, resulting in a decreased recall score. For example, compared to "imprisonment for more than 2 but less than 3 years," "imprisonment for more than 1 but less than 3 years'' would incur a lower penalty factor. We hope that the predicted imprisonment can be sufficiently precise on the range. Our aspiration is that the projected duration of imprisonment remains highly accurate within its intended range.
> Without the penalty factor, Eq. 1 in line 360 will degrade into a classical Recall score, which can only evaluate whether the predicted penalty type is correct or not, but unable to evaluate if the predicted imprisonment is ambiguous.
>
> &nbsp;
>
> **Question B:**
>
>  Figure 2: there is no baseline or previous SOTA to compare to. What score would a majority baseline get?
>
> **Response:**
>
> **Our goal is to comprehensively assess the influence of syllogism on State-of-the-Art (SOTA) LLMs for legal tasks.**
>
> As demonstrated in many recent papers (e.g., [5-7]), BERT-based models have exhibited a remarkable proficiency in surpassing majority baselines when tackling legal tasks within the scope of legal tasks. This pronounced advantage primarily stems from their robust language modeling capabilities. The escalation towards more potent Large Language Models (LLMs) as SOTA benchmarks have become an undeniable trajectory in recent research. In alignment with this trajectory, we have meticulously assembled an exhaustive compilation of cutting-edge LLMs, drawing from our best available knowledge. These curated models will serve as the cornerstone for our benchmarking standards. This strategic approach effectively eliminates the need for relying on conventional majority baselines, paving the way for the integration of advanced models into the framework.
>
> &nbsp;
>
> **Question C:**
>
> The ethics statement says that sensitive information has been anonymized by the authors (Line 590). How was that done?
>
> **Response:**
>
> Our proposed benchmark dataset is constructed based on CAIL-Long [8], in which a majority of cases have undergone anonymization to safeguard sensitive information.
>
> &nbsp;
>
> **Question D:**
>
> How were the LLMs prompted? Was there any randomness in the generation? This is important for reproducibility.
>
> **Response:**
>
> **We design specific prompts for each subtask.**
>
> Table 5 in the Appendix A demonstrates the prompt templates. The facts serve as input for the AR and CEG tasks, while AIG utilizes the outputs from AR and CEG as its input. Subsequently, LJP takes the output of AIG as its input.
>
> **There is no randomness in the generation.**
>
> We set the hyper-parameter temperature=0 to control the randomness of generation. The temperature parameter is used to adjust the probabilities of the predicted words in the softmax output layer of the model to control the randomness. Lower temperature indicates less random outputs.
>
> &nbsp;
>
> **Typos Grammar Style And Presentation Improvements:**
>
> 1. The colors in Figures 2 and 3 could be more color-blind friendly. See https://www.extensis.com/hs-fs/hubfs/Colorblind-Color-Palette.jpg?width=15003&name=Colorblind-Color-Palette.jpg for ideas.
> 2. The related work around Lines 147-163 enumerates papers and summarizes them. It would be more helpful to provide an overview, e.g. saying that many papers have addressed the task of LJP, and then cite a string of papers without going into too much detail.
> 3. Line 363 "scope of imprisonment" Do you mean "duration of imprisonment"?
> 4. "plaintext": this word is used multiple times, for example on Line 390. From the paper, I'm guessing that it means "without syllogism", i.e. prompting a model to predict charges without prompting it for the intermediate tasks. To help the reader understand, it would be good to clarify this, for example with a footnote.
>
> **Response:**
>
> Thanks for providing a lot of valuable suggestions for the details of the paper writing.
>
> 1. Thanks for your suggestion, we will change the colors of Figure 2 and 3 to be more color-blind friendly.
> 2. We will revise the description of related works as follows: With the development of neural networks and transformer-based models, there are some works use the neural networks to predict judgment results, such as, RNN-based models [9,10], BERT [9], reinforcement learning [11] and contrastive learning methods [12].
> 3. The “scope of imprisonment” means the range of possible imprisonment, for example, the scope of imprisonment is more than two but less than three years. While “duration of imprisonment” indicates the specific length of imprisonment.
> 4. Thanks for your suggestion, we will clarify that “plaintext” means “without syllogism” in the footnote.
>
> &nbsp;
>
> **References:**
>
> [1] Herbert Lionel Adolphus Hart et al. 2012. The concept of law.
>
> [2] Mingxuan Gao. 2009. Criminal Law.
>
> [3] Cong Jiang, and Xiaolei Yang, 2023. Legal Syllogism Prompting: Teaching Large Language Models for Legal Judgment Prediction. ICAIL.
>
> [4] Jason Wei et al. 2023. Chain-of-Thought Prompting Elicits Reasoning in Large Language Models. NeurIPS.
>
> [5] Junyun Cui et al. 2022. A Survey on Legal Judgment Prediction: Datasets, Metrics, Models and Challenges. Arxiv.
>
> [6] Gil Semo et al. 2022. ClassActionPrediction: A Challenging Benchmark for Legal Judgment Prediction of Class Action Cases in the US. ACL.
>
> [7] Ashutosh Modi. 2023. SemEval 2023 Task 6: LegalEval - Understanding Legal Texts. ACL.
>
> [8] Chaojun Xiao et al. 2021. Lawformer: A Pre-trained Language Model for Chinese Legal Long Documents. Arxiv.
>
> [9] Ilias Chalkidis et al. 2019. Neural Legal Judgment Prediction in English. ACL.
>
> [10] Wenmian Yang et al. 2019. Legal judgment prediction via multi-perspective bi-feedback network. IJCAI.
>
> [11] Jie Zhao et al. 2022. Charge prediction by constitutive elements matching of crimes. IJCAI.
>
> [12] Han Zhang et al. 2023. Contrastive learning for legal judgment prediction. TOIS.

---

### Meta-Review · Area_Chair_WbSy · 2023-09-17

**Recommendation:** 5

**Metareview:**

The paper presents a structured approach to Legal Judgment Prediction, emphasizing interpretability and explainability. It introduces a new dataset, SLJA, based on the concept of syllogistic reasoning. This dataset encompasses four tasks: article retrieval (AR), criminal element generation (CEG), article interpretation generation (AIG), and legal judgment prediction (LJP). The data was annotated using predictions from ChatGPT, which were then post-edited by crowdworkers. The paper evaluates the performance of various Large Language Models (LLMs) on this dataset, aiming to provide an in-depth analysis of their capabilities in legal judgment tasks.

The paper offers a systematic method to incorporate interpretability into AI-driven legal decisions. The introduction of the SLJA benchmark dataset is a significant contribution, and the annotation process employed is robust. The dataset's value is further enhanced by human corrections, ensuring its reliability. The paper is well-written, making it easily comprehensible. The use of LLMs for legal judgment analysis is timely and necessary, and the paper's syllogistic reasoning approach provides a novel perspective on legal judgment analysis. The evaluation of multiple LLMs on legal tasks further underscores the paper's significance.

The experimental setup is limited, relying solely on LLMs, which may not fully demonstrate the dataset's value. The use of LLMs, especially for tasks with multiple labels like LJP, raises concerns due to their unpredictability and slower inference speeds. Existing state-of-the-art LJP models have reportedly achieved better performance, making the choice of LLMs questionable. The paper could benefit from exploring other dominant theories for analyzing legal judgment, like the two-stage system, and expanding its scope beyond criminal law. Comparisons with classical methods and LLMs fine-tuned with legal data could provide a more comprehensive evaluation.

---

### Decision · Program_Chairs · 2023-10-07

**Decision:**

Accept-Main

**Comment:**

The paper presents a structured approach to Legal Judgment Prediction, emphasizing interpretability and explainability. It introduces a new dataset, SLJA, based on the concept of syllogistic reasoning. This dataset encompasses four tasks: article retrieval (AR), criminal element generation (CEG), article interpretation generation (AIG), and legal judgment prediction (LJP). The data was annotated using predictions from ChatGPT, which were then post-edited by crowdworkers. The paper evaluates the performance of various Large Language Models (LLMs) on this dataset, aiming to provide an in-depth analysis of their capabilities in legal judgment tasks.

The paper offers a systematic method to incorporate interpretability into AI-driven legal decisions. The introduction of the SLJA benchmark dataset is a significant contribution, and the annotation process employed is robust. The dataset's value is further enhanced by human corrections, ensuring its reliability. The paper is well-written, making it easily comprehensible. The use of LLMs for legal judgment analysis is timely and necessary, and the paper's syllogistic reasoning approach provides a novel perspective on legal judgment analysis. The evaluation of multiple LLMs on legal tasks further underscores the paper's significance.

The experimental setup is limited, relying solely on LLMs, which may not fully demonstrate the dataset's value. The use of LLMs, especially for tasks with multiple labels like LJP, raises concerns due to their unpredictability and slower inference speeds. Existing state-of-the-art LJP models have reportedly achieved better performance, making the choice of LLMs questionable. The paper could benefit from exploring other dominant theories for analyzing legal judgment, like the two-stage system, and expanding its scope beyond criminal law. Comparisons with classical methods and LLMs fine-tuned with legal data could provide a more comprehensive evaluation.